# Crack Segmentation on Earthen Heritage Site Surfaces

**Yuan Zhang ***, **Zhiyong Zhang ***, **Wu Zhao and Qiang Li**

School of Information Science and Technology, Northwest University, Xi'an 710127, China
* Correspondence: zhangyuan@nwu.edu.cn (Y.Z.); zhangzy@nwu.edu.cn (Z.Z.)

**Abstract:** Earthen heritage sites are historical relics left by ancient human activity, with earthen as the primary building material, and have significant historical, scientific, and artistic value. However, many sites have experienced extensive deterioration caused by environmental forces and human factors. A crack is a kind of typical damage to the walls of earthen heritage sites. Studies of the crack-formation process can effectively predict trends in damage, which will play a critical role in the maintenance of earthen heritage sites. This study is the first of its kind to propose a deep learning method to study the cracks on earthen heritage sites at the pixel-level, adopt the idea of transfer learning, and employ a mixed-crack image dataset for training three deep learning models. The precision, recall, IoU, and F1 metrics were used to evaluate the performance of the trained models. The experimental results showed that FPN-vgg16 appeared to have the highest level of applicability to detect cracks on earthen heritage sites among all networks, due to the highest F1 score of 84.40% and the highest IoU score of 73.11%. The results illustrated that the proposed method in this paper can effectively be used to analyze the rammed earth surface crack images, with great potential in related research fields.

**Keywords:** crack detection; pixel-level segmentation; classification; deep learning

## 1. Introduction

Earthen heritage sites are historical relics with earthen as the main building material, left by ancient human activities, which are typically found in dryland regions because of the wide availability of earthen materials and favorable environmental conditions. Rammed earth has been widely used to build dwellings and temples in ancient Europe and Asia. In China, these earthen heritage sites are relatively concentrated in the arid regions of Northwest China, mainly including ancient cities, the Great Wall, passes, beacon towers, earthen pagodas, and tombs. Earthen heritage constitutes 10% of sites on the World Heritage List [1] and has important historical, scientific, and artistic value. However, many sites have experienced extensive deterioration mediated by environmental and human activities, which threatens to reduce their value [2,3]. Many heritage sites composed of adobe materials the earthen structures with significant value remain in the Gobi Desert and other desert regions in Northwest China [4]. Suoyang Ancient City is a typical rammed earth heritage site, which was built during the Han (206BCE–220CE) and Tang (618–907CE) dynasties [5]. It is located in Gansu Province, northwest China, along the ancient Silk Road (Figure 1a). This ancient city site, situated in arid areas, has all kinds of harsh environmental conditions, such as strong wind, sandstorms, and drought. The damage to the walls of earthen heritage sites caused by the harsh environment include scouring, flaking, and cracking (Figure 1c–e). Many investigations have been conducted at the Suoyang Ancient City site and some typical damage was spotted. This study aimed to find a long-term method for researching the deterioration of earthen heritage sites by studying the cracks on the rammed earth surfaces of the sites.

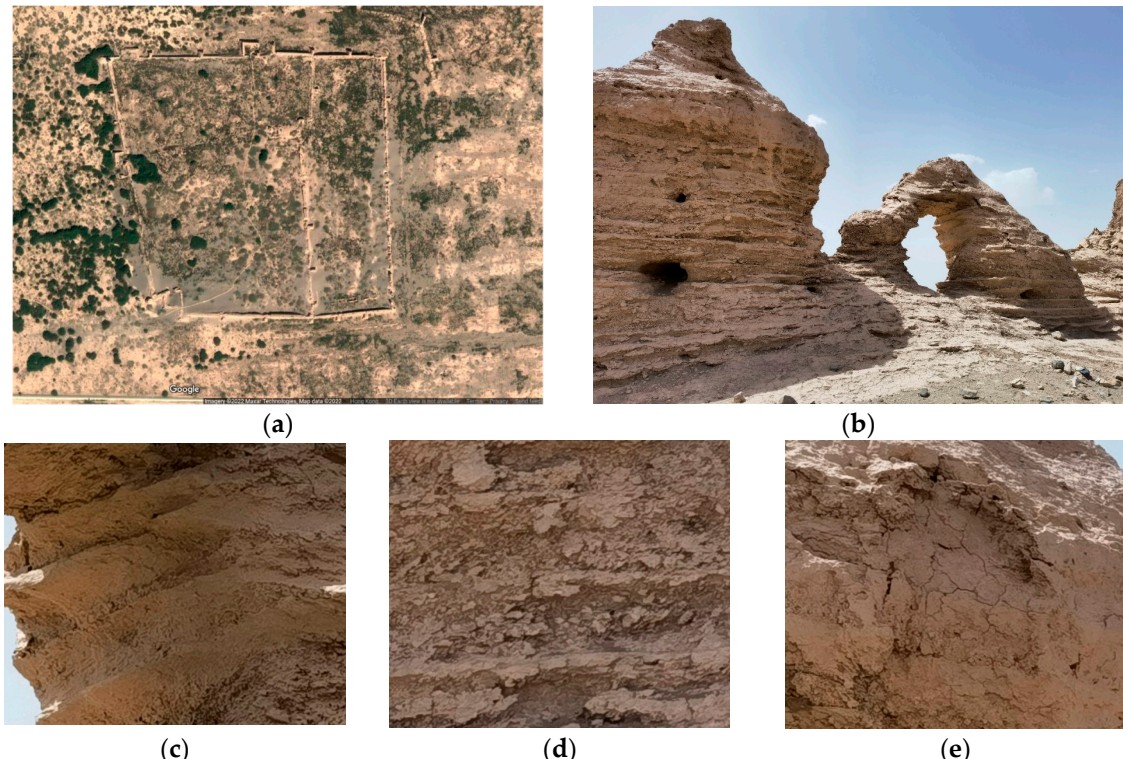

**Figure 1.** Suoyang Ancient City. (**a**) Google Maps image of Suoyang Ancient City. The photos show (**b**) some typical damage to wall surfaces, including (**c**) scouring, (**d**) flaking, and (**e**) cracking.

Cracks comprise the damage to the main structures of earthen heritage sites. Detecting the crack formation process can effectively predict the trend in the damage to earthen sites and thus help officials adopt appropriate methods to maintain the sites' value. The traditional method of crack detection was measuring the length and width of the cracks through visual inspection by well-trained cultural relic workers with the help of rulers. Although the current state of the crack can be recorded via manual inspection, long-term observation of the crack and the objectivity and reliability of the crack observation cannot be effectively guaranteed. The replacement of observers will lead to the instability of observations. Therefore, many researchers focus on developing automated and efficient crack-detection techniques for the long-term monitoring of earthen heritage sites.

The application of computer vision-based crack detection technology for modern buildings, such as cement roads, brick walls, and bridges, has been widely studied. Threshold segmentation [6,7], morphology [8], wavelet transform [9], and filter-based algorithms [7,10,11] have been applied in the field of crack detection. Wang et al. [12] summarized traditional computer vision-based crack detection and subdivided them into four categories, namely an integrated algorithm, morphological approach, percolation-based method, and practical technique, and then analyzed and compared the experimental results of the four types of methods. Using these methods to analyze a photo of the building surface, one can identify the cracks, locate the cracks, and measure the cracks' width, length, and orientation. By adjusting the parameters of these algorithms, high inspection detection accuracy can be achieved. The traditional computer vision-based crack detection algorithm usually focuses on the understanding and analysis of the partitive features of the image, and parameters are designed for a specific dataset. However, when there is interference, such as stains, potholes, etc., the partitive image will be difficult to segment. The main characteristics of rammed earth heritage are that cracks are distributed on the irregular surface and mixed with irregular textures, such as shadows and holes. It is difficult for computer vision methods to segment cracks.

In order to overcome the drawbacks of traditional computer vision-based algorithms, some studies have suggested applying machine learning to image processing tasks. Lecun et al. first proposed that Convolutional Neural Networks (CNN) could be applied to handwritten character recognition [13]. More studies have proposed image classification [14–16], object detection [17–19], semantic segmentation [20–22], and crack detection [23–25] tasks. Instead of extracting features, Convolutional Neural Networks can automatically learn the features of an image and identify cracks from images. Deep learning semantic segmentation models have been extensively studied for crack detection in modern buildings with positive results.

Classical machine learning algorithms, such as Support Vector Machines [26] and Random Forests [27], require the manual extraction of crack features. Even though this sort of algorithm relies heavily on manual extraction, they produce some fruitful study findings. The accuracy of the algorithm will be diminished to varying degrees due to the variations in external environmental factors, such as background noises and lighting conditions, when extracting manually. These factors may lead to unexpected practical problems and limitations in the application of classical machine learning algorithms.

As a branch of machine learning, the architecture of deep learning can extract features related to it and thus can effectively avoid the limitations existing in traditional algorithms. Research on crack detection technology based on deep learning, including image classification, object recognition, and semantic segmentation, has been proposed. Image classification algorithms can automatically classify the images into two categories, crack images and non-crack images. Object recognition algorithms can find and locate objects that contain cracks in images and use bounding boxes to identify and locate them.

Semantic segmentation is basic research in the field of computer vision processing and has been widely used in many fields, such as autonomous driving, medical image analysis, etc. Compared with image classification and object recognition, semantic segmentation is more accurate in positioning, and its positioning output can be distinguished using different colors. It can identify the width and length of cracks at the pixel-level. Obviously, in crack detection for earthen heritage sites, it is important not only to detect whether there are cracks on the surface but also determine the size of the cracks, which is of great significance in studying the causes and development of the cracks and proposing scientific protection plans.

Ni et al. [28] used GoogLeNet and ResNet classification architecture to find crack patches, employed Otsu's thresholding to segment crack patches, and adopted median filtering to eliminate the influence. Kang et al. [29] proposed the faster region proposal convolutional neural network (Faster R-CNN) algorithm to detect crack regions. Wei et al. [30] used Mask R-CNN to perform crack segmentation at the pixel-level. Several studies have employed two settings: first, the classification architecture was used to locate crack patches, and then, the crack map was extracted at the pixel-level. The approach above is known as Hybrid Semantic Segmentation.

Another method was performed with encoder-decoder structures without identifying a crack patch or crack region beforehand. A typical encoder-decoder structure consists of an encoder network and a corresponding decoder network. The encoder network has a backbone architecture that contains convolution, pooling, and activation layers. The role of the decoder network is to map the low-resolution feature maps to full-resolution feature maps for pixel-wise classification.

Several crack segmentation methods with encoder-decoder structures have been developed to detect surface cracks, such as U-Net [31], SegNet [32,33], LinkNet [34], ResNet [35], and several variants [36,37]. In the semantic segmentation setting, the key to achieving improved accuracy is to provide the contextual information flow in architecture [38].

U-Net was initially released in the context of medical image analysis for biomedical image segmentation [31]. In several studies, the U-Net architecture has been proposed for crack detection. Sizyakin et al. [39] used morphological filtering to improve the binary map with localized cracks from the U-Net neural network. Lau et al. [40] put forward a U-Net-based architecture that replaces the encoder with a pre-trained ResNet-34 backbone.

Lin et al. [41] advanced a U-Net architecture that employs the Full Attention Strategy, which comprises synthesis of the attention mechanism and the outputs from each encoding layer in a skip connection. Cao et al. [42] used the squeeze-and-excite module to empower U-Net, which they call SE-U-Net. This architecture can adaptively learn the shallow information matched with deep features in the skip path. The preceding studies demonstrated its superiority over the methods based on traditional machine learning algorithms.

SegNet [32] is made up of an encoder network and a corresponding decoder network. The encoder network architecture is topologically identical to the VGG16 [43] network with 13 convolutional layers. Song et al. [44] deployed SegNet architecture to segment structure surface cracks. By using SegNet as a reference, Chen et al. [45] recommended an encoder-decoder structural model for inspecting concrete pavement, asphalt pavement, and bridge deck cracks. Zou et al. [46] proposed the DeepCrack network based on the encoder-decoder architecture of SegNet and pairwisely fused the convolutional features that are generated in the encoder network and decoder networks. LinkNet employs an improved codec network architecture, and the input of each encoder layer is also bypassed by the output of its corresponding decoder [34]. This method can recover the spatial information lost by downsampling operations in the encoder and reduces the processing time. Training deeper neural networks is more challenging. He et al. presented a residual learning framework to make it simpler to train networks [35]. ResNet, which has a depth of up to 152 layers, is less complex than VGG. Combining the Residual connections and Inception architecture [47], Szegedy et al. [36] proposed that Inception-ResNet improves training speed with low total parameter counts and computational costs. Loverdos et al. [48] evaluated the effectiveness of several deep learning networks (U-Net, DeepLabV3+, U-Net (SM), LinkNet (SM), and FPN (SM)) in brick segmentation and crack detection with masonry walls. The analysis results showed that machine learning is superior to image-processing techniques for segmentation. Deep learning demonstrates enormous potential in crack detection.

However, there is not much research being done on the application of convolutional neural network models for the detection of cracks in earthen heritage sites because deep learning models require huge amounts of training data to make them work, and the adoption of deep learning techniques relies significantly on the volume of data available. The complexity of building a deep learning model stems from the difficulty in obtaining a significant number of cracked photographs of earthen heritage sites.

This study focused on contributing to this growing area of research by exploring an effective and efficient method for crack detection in earthen heritage sites. By evaluating different network performances, pixel-level crack segmentation models were recommended for the study of earthen heritage site surfaces. It also aimed to establish a hybrid dataset. Different open-source image datasets were mixed, and then, a hybrid dataset was constructed. The hybrid dataset combines the images of cracks on the surface of cement pavement, asphalt pavement, and earthen heritage sites.

In this paper, a method for crack detection in earthen heritage sites is proposed. Mixed datasets were used to evaluate different network performances by using key metrics and manually labeled ground truth for evaluation purposes.

The rest of the paper is structured as follows. Section 2 briefly presents the mixed datasets and proposed network. Section 3 presents experimental results. The results are analyzed and discussed in Section 4, and Section 5 draws some conclusions and presents some hints for future work.

## 2. Materials and Methods

### 2.1. Overview

This study evaluated the reliability and precision of the application of different methods to cases of cracks in the earthen surfaces of the Suoyang Ancient City and proposed a deep learning method that can effectively detect and predict rammed earthen heritage cracks. The workflow of the proposed method approach is illustrated in Figure 2.

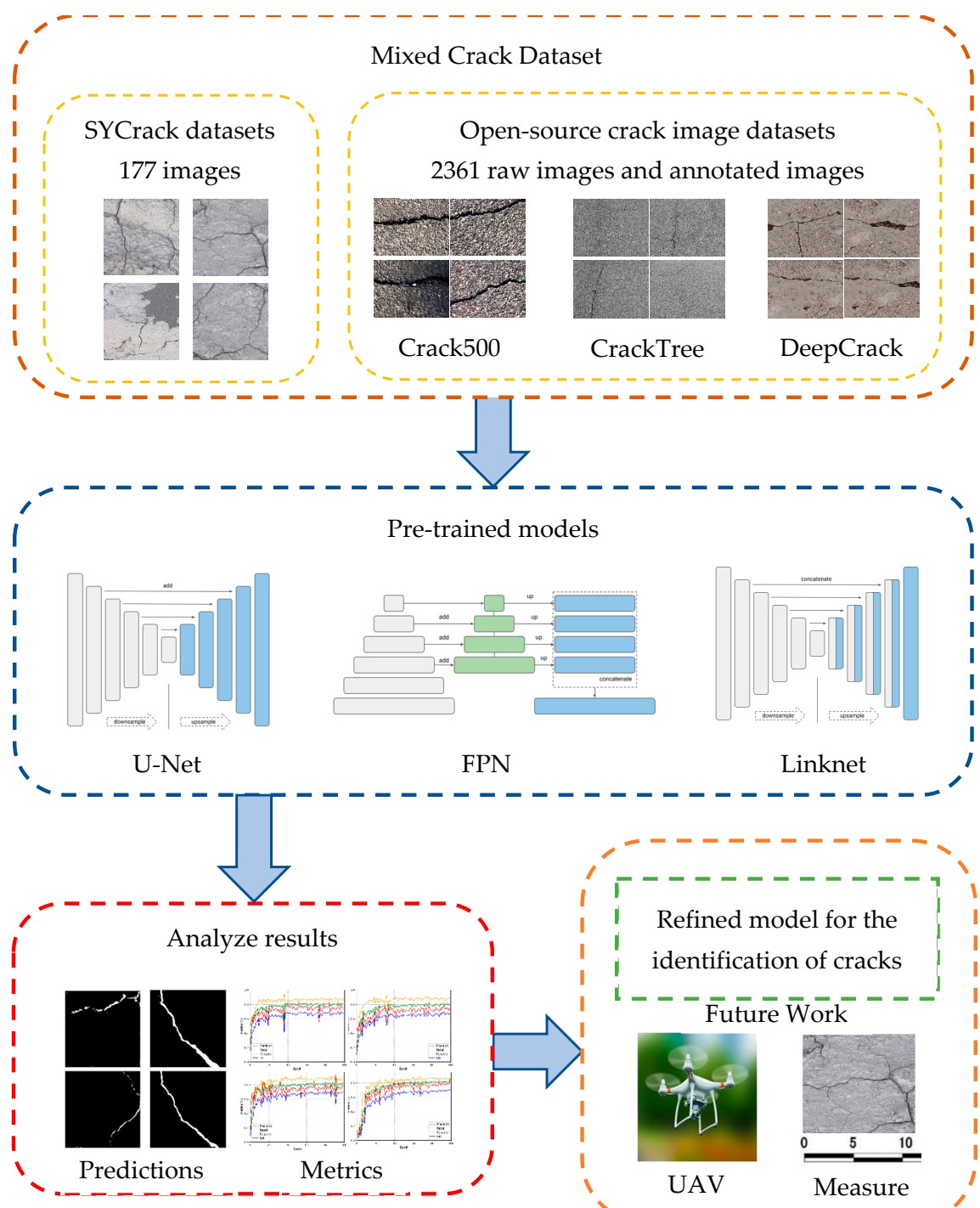

**Figure 2.** Workflow of the proposed method for detecting and predicting cracks in rammed earthen heritage sites.

Step1: Gather open-source crack image datasets, which include road cracks with occlusions and shadow interference. Every image has a manually annotated binary ground truth image.

Step2: Take some photographs of the crack on the surface of the Suoyang Ancient City wall. These photos were cropped to a uniform size, and 177 photos were selected and manually annotated to generate a dataset.

Step3: Combine the open-source crack dataset and the Suoyang Ancient City crack dataset, training 3 models with the mixed crack dataset to optimize model parameters.

Step4: Analyze results using key metrics and predictions, devoted to the segmentation of cracks on the surface of earthen heritage sites.

### 2.2. Image Dataset

Many open-source crack image datasets were widely used, such as CrackTree206 [49], Crack500 [50], and DeepCrack [51], etc. The CrackTree206 dataset contains 206 $800 \times 600$ pixel photos of road cracks, with occlusions and shadow interference. It was widely used because it was released early with many samples. The Crack500 dataset contains 500 images with a resolution of around $2000 \times 1500$. Each image is cropped into 16 non-overlapping image regions and saved as 3368 crack images, with 1896 images in the training data, 1124 images in the testing data, and 348 images in the validation data. The DeepCrack dataset contains 537 crack images with three textures and two scenes, where the training data contains 300 images and the test data contains 237 images. All above datasets were manually annotated and generated binary ground truth images.

We collected many images of the cracks on the Suoyang Ancient City wall surface and removed some images with excessive interference. A dataset, named SYCrack, including 177 images with a uniform size of $256 \times 256$ pixels images, was generated. This dataset contains two categories: images with cracks and images without cracks. All of these images were manually annotated and rendered as binary ground truth images. Then, 80% of these images were chosen as training datasets, with 20% as testing datasets.

To improve the robustness of models, CrackTree206, Crack500, Deep Crack, and SYCrack were merged to form a Mixed Crack Dataset (MCD). The MCD, with a total of 2538 raw images and annotated images, contains concrete cracks, asphalt cracks, and earthen cracks.

### 2.3. Proposed Network

The semantic segmentation network consists of an encoder and a decoder, and different pre-trained Convolutional Neural Network architectures are used as backbones. As stated before, related works show that U-net, FCN, and Linknet have outstanding performance in image segmentation, especially in detecting cracks in some specific buildings. Their application, however, is limited only to images of those buildings. A deep learning network that was trained on a specific material dataset cannot achieve the same accuracy with different materials [52]. This means that the model trained on the cement road datasets is not necessarily suitable for the cracks in walls of earthen heritage sites. Özgenel et al. [53] proposed that a pre-trained CNN network can extract the low-level features of cracks. Therefore, with only the requirement for a small number of training samples and fast convergence networks, models can be achieved for new materials.

### 2.4. Transfer Learning

Deep learning requires enough data to complete the model training. The current dataset of earthen cracks cannot meet the needs of model training, and thus, the transfer learning method was adopted.

Transfer learning involves making the best possible use of the knowledge in the annotated domain to find the similarity of the target problem to assist in knowledge acquisition and learning in the target field. The transfer learning task, based on similarity, applies the model trained in the source domain to the target domain. Transfer learning can solve the insufficiency of high-quality training data in the target domain, and even in the case of a small amount of data, it can also solve the problem of model training.

In this research, we adopted the concept of transfer learning and used the road cracks similar to the surface cracks of earthen heritage sites to combine the pictures of the crack images of the earthen heritage sites for deep learning model training. This was verified using the Suoyang Ancient City crack image testing dataset to compare the deep learning models and select the best the crack identification model.

## 3. Verification and Results

### 3.1. Experimental Environment

All models were created in the Python 3.8 environment using the Tensorflow framework, and training and testing were carried out in the following experimental setting. Tables 1 and 2 detail the software and hardware environments, respectively.

**Table 1.** Software environment setups employed in this paper.

| Software | Details |
| --- | --- |
| OS | Windows 10 Professional |
| Programming language | Python 3.8 |
| Deep learning framework | Tensorflow-gpu 2.3.0 |
| Dependent library | Cuda, Keras, PIL etc. |

**Table 2.** Hardware environment setups employed in this paper.

| Hardware | Details |
| --- | --- |
| CPU | Intel(R) Xeon(R) CPU E5-2670 v3 @ 2.30GHz |
| CPU RAM(GB) | 256 |
| GPU | NVIDIA GeForce RTX 3070 |
| GPU memory (GB) | 8 |
| CUDA cores | 5888 |
| CUDA version | 11.2 |

### 3.2. Experimental Data

The open-source dataset and crack dataset of the Suoyang Ancient City were mixed. The mixed crack data contain 2538 raw images and annotated ground truth images. Figure 3 shows samples of the training dataset.

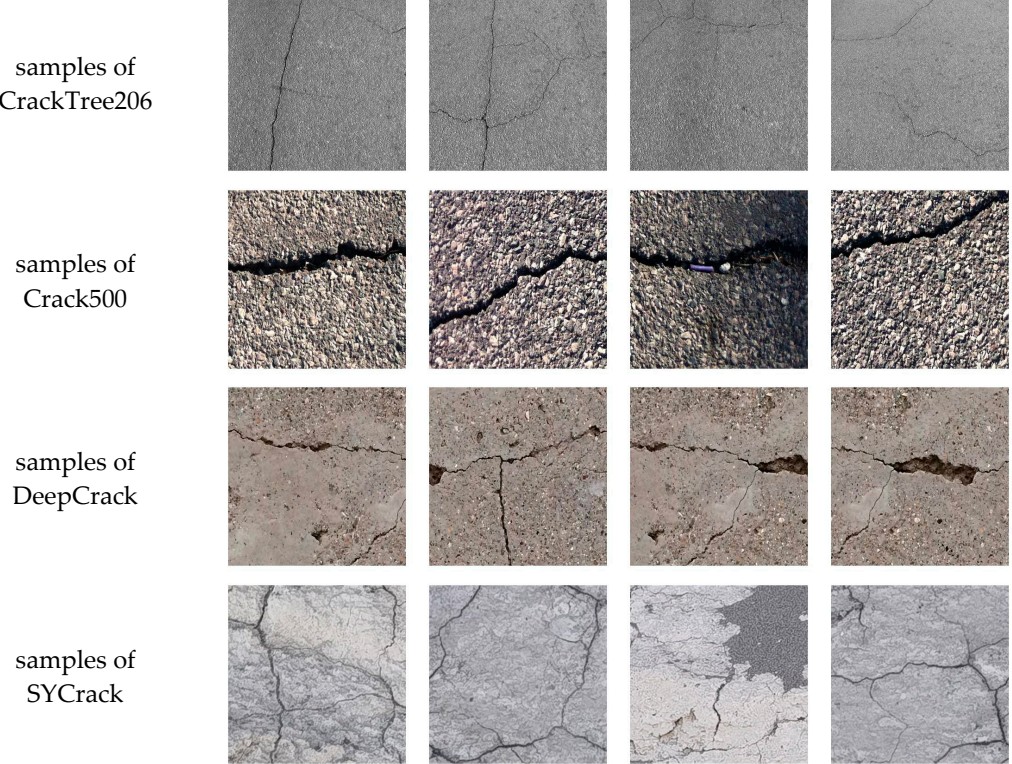

**Figure 3.** Samples of cracks in the training dataset.

### 3.3. Analysis of the Pre-Trained Deep Learning Models

The deep neural network model architectures used in this paper were Unet, Linknet, and FPN, with four backbones for each architecture, and all backbones have pre-trained weights on ImageNet [54]. The number of parameters is a performance indicator of the model, which means the requirement for computing resources. The larger number of parameters, the more storage resources and computing time required. Table 3 shows the statistics of the total number of parameters for each model.

**Table 3.** Total number of parameters for each model.

| Model | Backbones | Parameters |
|---|---|---|
| Unet | vgg16 | 23,752,273 |
| | resnet152 | 67,295,194 |
| | densenet201 | 26,378,577 |
| | inceptionv3 | 29,933,105 |
| Linknet | vgg16 | 20,325,137 |
| | resnet152 | 63,517,466 |
| | densenet201 | 22,549,393 |
| | inceptionv3 | 26,268,401 |
| FPN | vgg16 | 17,572,545 |
| | resnet152 | 61,645,898 |
| | densenet201 | 21,523,905 |
| | inceptionv3 | 25,029,281 |

As can be seen from Table 3, the parameters of FPN were the fewest, indicating a simple model structure and a lower memory footprint. The number of model parameters varied by more than three times due to different backbone networks.

### 3.4. Training Configuration

All models were trained for 100 epochs until the F1 score on the validation set stopped increasing. The data were supplied to the network with a batch size of 4. The optimization algorithm employed was Adam, which is efficient for gradient descent and requires less memory. The learning rate was set as 0.001. The Binary Cross-Entropy loss function (see Equation (1)) was implemented here, as the function works well for classification objectives, such as segmentation at the pixel-level [55]. Binary Cross-Entropy is defined as:

$$L_{BCE}(y, \hat{y}) = -(y log(\hat{y}) + (1 - y)log(1 - \hat{y})) \tag{1}$$

where $y$ is the ground truth and $\hat{y}$ is the prediction. Additionally, $y$ can have values of 0 (background) or 1 (crack), whereas $\hat{y}$ can have values between 0 and 1.

The evaluation of the network is based on the values of *Precision*, *Recall*, *IoU*, and *F1 score*. These metrics are given as:

$$Precision = \frac{TP}{TP + FP} \tag{2}$$

$$Recall = \frac{TP}{TP + FN} \tag{3}$$

$$F1\ score = \frac{2 \times Precision \times Recall}{Precision + Recall} \tag{4}$$

$$IoU = \frac{GroundTruth \cap Prediction}{GroundTruth \cup Prediction} \tag{5}$$

The *TP*, *FP*, *TN*, and *FN* terms are defined as follows:

*True Positive* (*TP*): the pixel is a crack (Positive) and is identified as a crack (True);
*False Positive* (*FP*): the pixel is intact (Positive) and is identified as a crack (False);
*True Negative* (*TN*): the pixel is intact (Negative) and is identified as intact (True);

*False Negative (FN)*: the pixel is a crack (Negative) and is identified as intact (False).

Precision and Recall are the two most common metrics. Additionally, they form the basis of the F1 score. Precision is the percentage of correct positive predictions versus the total number of positive predictions. Recall indicates the completeness of the positive predictions. Normally, poor precision is caused by high recall scores and vice versa. The F1 score combines the accuracy and recall metrics and is defined as the harmonic mean of precision and recall.

The F1 score is a useful model performance metric when working on classification models with an imbalanced dataset. The intersection over Union (IoU) is a metric used to measure the accuracy of an annotation based on a task. The F1 score and IoU score were used as the main indicators to evaluate the model.

## 4. Results and Discussion

All models were trained for 100 epochs with a batch size of 4 for all models. Table 4 shows the metrics presented for the validation set.

**Table 4.** The metrics are presented for the validation set.

| Model | Backbones | Precision% | Recall% | F1 Score% | IoU% | Best Epoch * |
|---|---|---|---|---|---|---|
| Unet | vgg16 | 89.41 | 85.52 | 83.65 | 72.00 | 84 |
| | resnet152 | 93.02 | 79.85 | 82.28 | 69.98 | 93 |
| | densenet201 | 92.60 | 94.38 | 83.68 | 72.01 | 80 |
| | inceptionv3 | 95.21 | 80.66 | 82.65 | 70.51 | 84 |
| Linknet | vgg16 | 90.19 | 85.97 | 82.76 | 70.74 | 99 |
| | resnet152 | 90.40 | 83.07 | 82.63 | 70.43 | 76 |
| | densenet201 | 89.80 | 81.56 | 81.83 | 69.33 | 96 |
| | inceptionv3 | 88.03 | 80.20 | 80.88 | 67.99 | 88 |
| FPN | vgg16 | 88.83 | 85.82 | 84.40 | 73.11 | 93 |
| | resnet152 | 90.50 | 83.35 | 83.31 | 71.46 | 89 |
| | densenet201 | 87.84 | 82.62 | 82.59 | 70.44 | 90 |
| | inceptionv3 | 90.05 | 85.37 | 81.72 | 69.25 | 94 |

* Epoch where the highest F1 score was obtained for the validation set.

Metrics of U-Net, As shown in Figure 4, Precision achieved the highest score at epoch 36, but the Recall score was only 40.38%, which means more cracked pixels were classified as intact. After completing training based on 100 epochs, the Precision score was 88.08%, but the Recall value was only 74.59%. Metrics of Linknet and FPN were also analyzed, shown in Figures 5 and 6, revealing similar results. Furthermore, the authors checked the predicted outputs, as shown in Figure 7, the crack annotation was incomplete for the predicted result and there was noise in the background. In other words, neither the Precision nor Recall score achieved good results. Moreover, the IoU value was 67.43%, which also shows that the prediction result did not match the ground truth very well. Among the results predicted by U-Net-densenet201, U-Net-resnet152, FPN-resnet152, and FPN-inceptionv3 models, some cracked pixels were classified as intact, which indicates failure in detecting some cracked pixels. In the results predicted by the Linknet-densenet201, Linknet-inceptionv3, FPN-densenet201 models, there was too much noise in the background.

U-Net-vgg16, Linknet-vgg16, and FPN-vgg16 all performed well without noise in the background. As can be seen in Figures 4–6, the results of completing training based on 100 epochs had very close F1 scores. According to Table 4, FPN-vgg16 obtained the highest F1 score of 84.40% and the highest IoU score of 73.11%, which indicates the highest level of applicability in crack detection of earthen heritage sites among all networks in this study. FPN-vgg16 demonstrated unexpected performance and has fewer training parameters than other networks when F1 and IoU scores are compared and the prediction result image is analyzed.

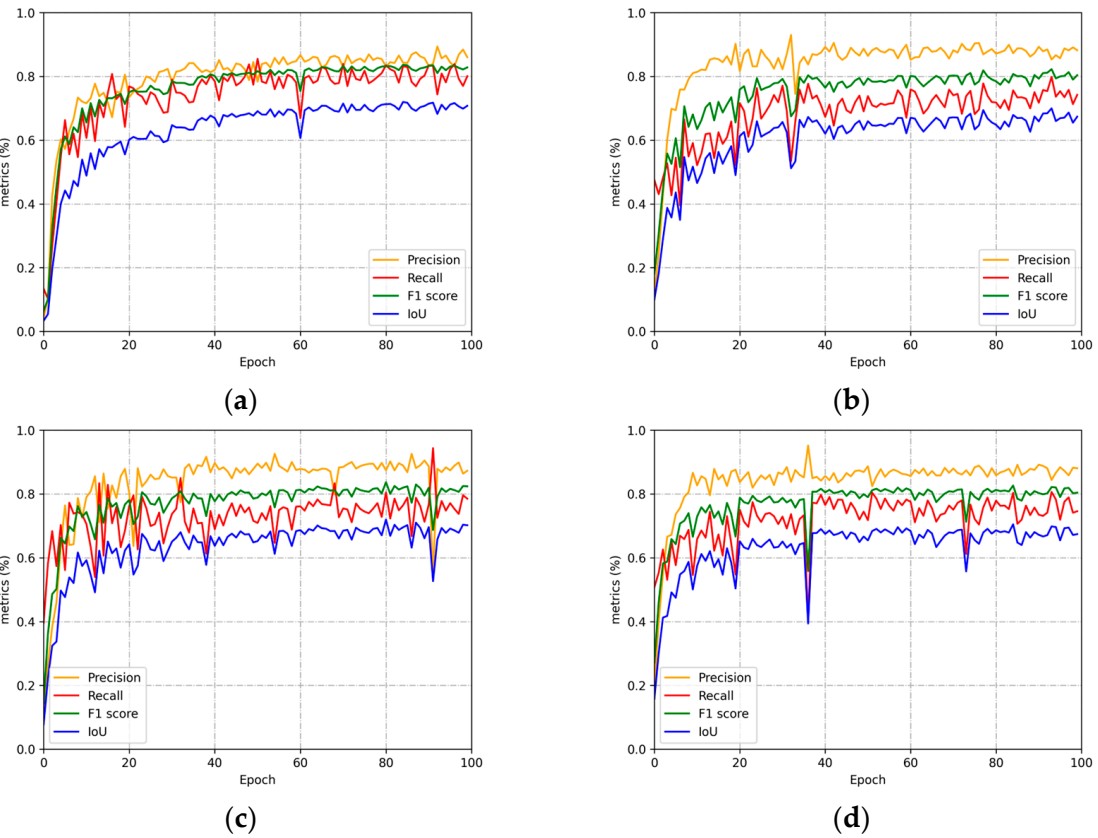

**Figure 4.** The metrics Precision, Recall, IoU, and F1 score as obtained from U-Net for different backbones: (**a**) vgg16, (**b**) resnet152, (**c**) densenet201, (**d**) inceptionv3.

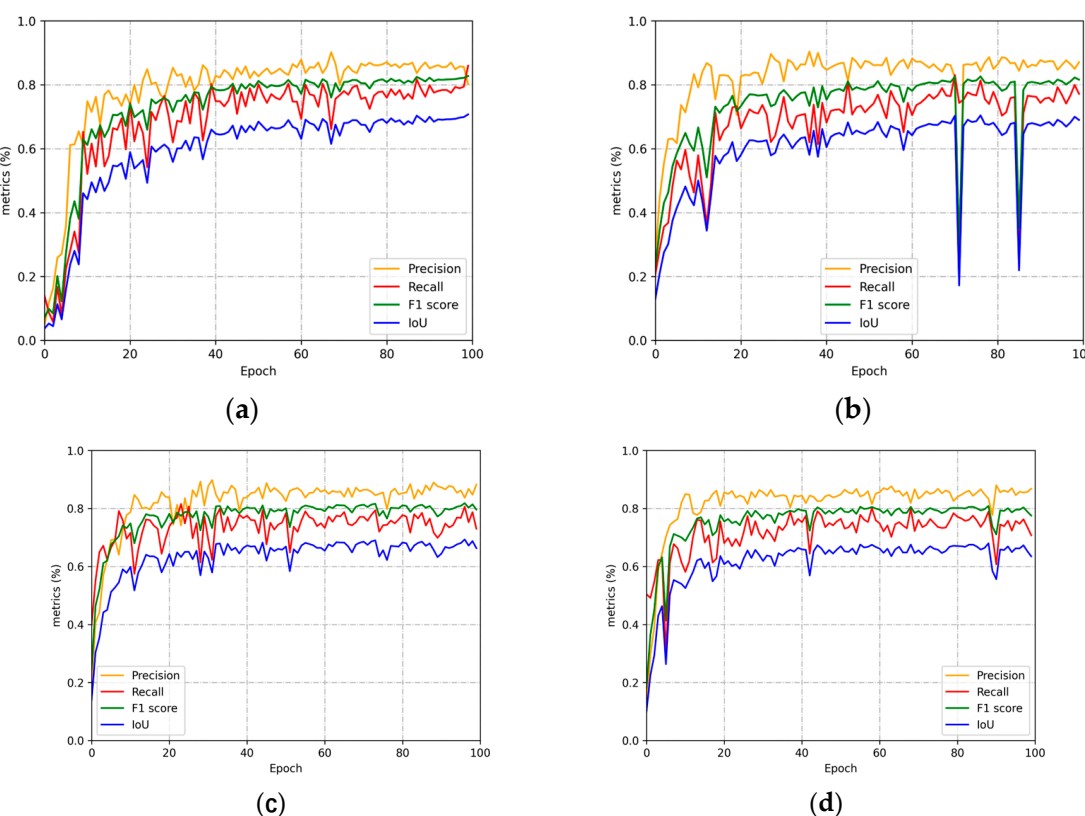

**Figure 5.** The metrics Precision, Recall, IoU, and F1 score as obtained from Linknet for different backbones: (**a**) vgg16, (**b**) resnet152, (**c**) densenet201, (**d**) inceptionv3.

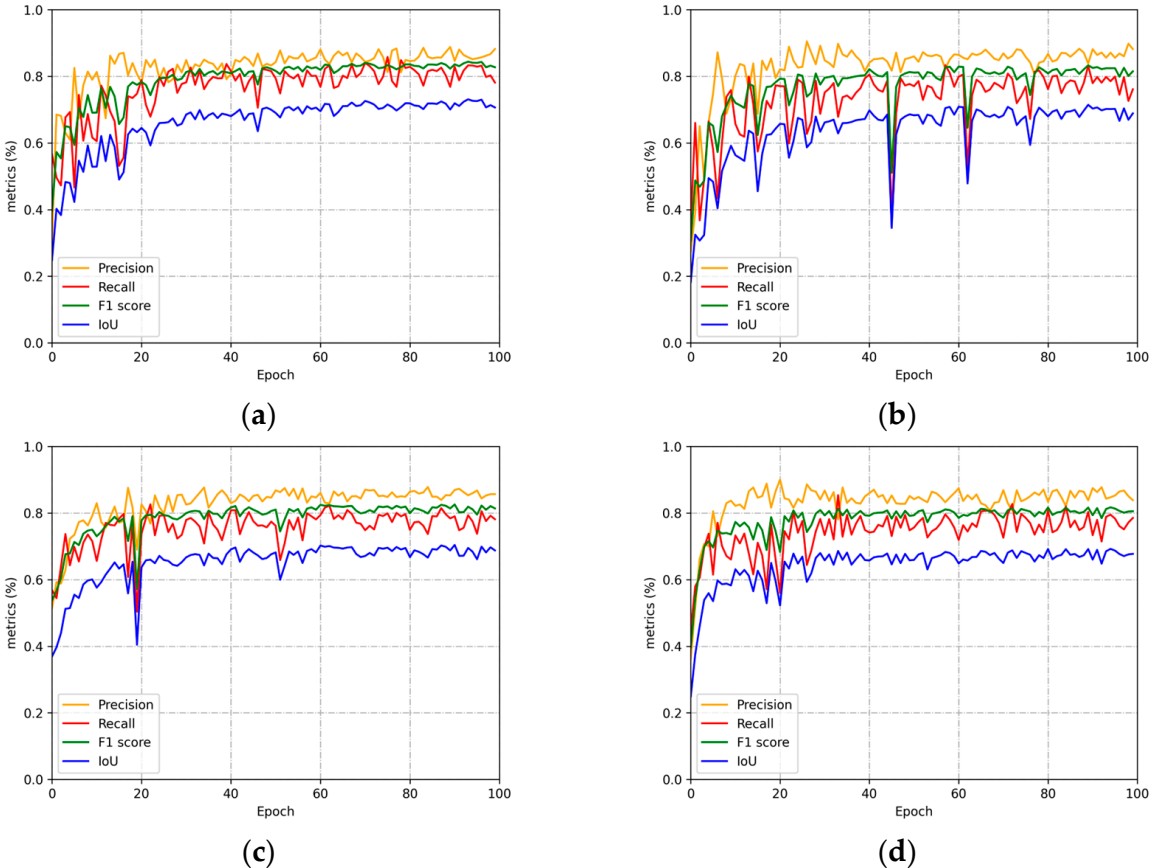

**Figure 6.** The metrics Precision, Recall, IoU, and F1 score as obtained from FPN for different backbones: (**a**) vgg16, (**b**) resnet152, (**c**) densenet201, (**d**) inceptionv3.

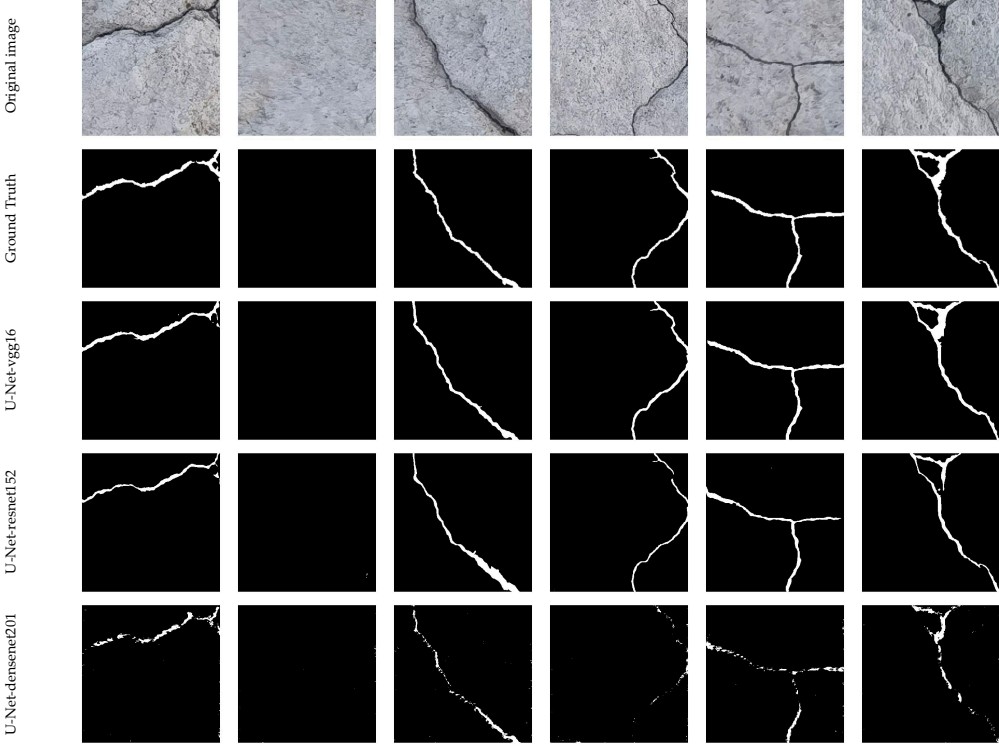

**Figure 7.** *Cont.*

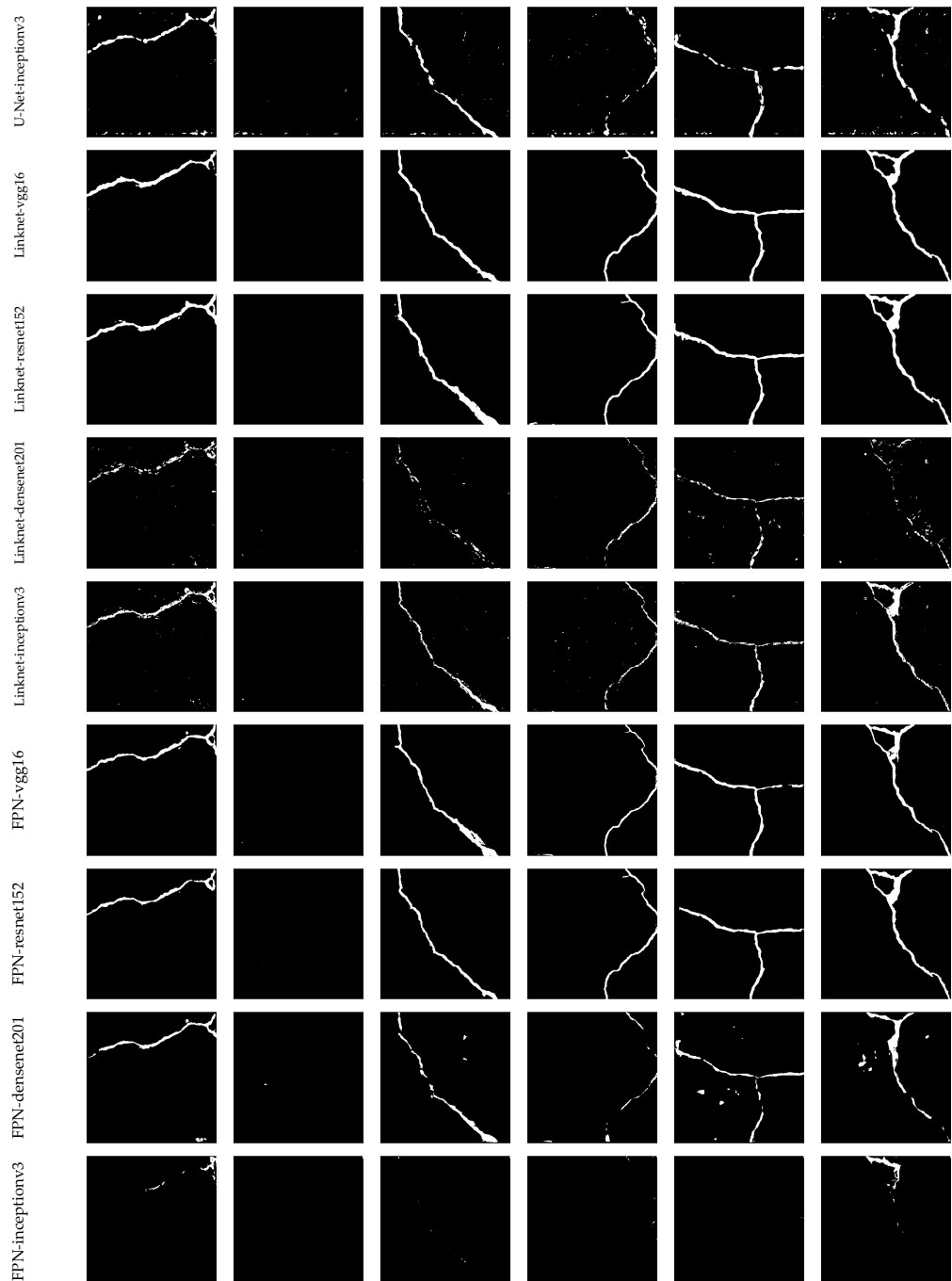

**Figure 7.** The original image, ground truth, and predictions with all models for different images from the validation dataset.

## 5. Conclusions and Future Work

### 5.1. Conclusions

This paper proposes a method for crack detection in earthen heritage sites by evaluating the performances of different networks. Pixel-level crack segmentation models were recommended to detect cracks on earthen heritage site surfaces. The three models evaluated in this study are the most promising crack detection networks, and the four backbones show different abilities in prediction. Through the comparison of F1 and IoU scores and the analysis of the prediction result picture, FPN-vgg16 shows surprising performance and has fewer training parameters than other networks.

We collected some crack pictures at the site of Suoyang City, and cropped, screened, and annotated 177 pictures of cracks. A hybrid dataset, a mixture of different opensource image datasets, was constructed, which combines the images of cracks on the surface of cement pavement, asphalt pavement, and earthen heritage site surfaces, which thus solves the problem of a lack of data on cracks on earthen heritage site surfaces. To the best of the authors' knowledge, this study is the first to propose pixel-level crack segmentation for surface cracks in earthen heritage site surfaces.

Monitoring the change in cracks on earthen heritage site surfaces is a long-term research topic. The crack detection method that we proposed can help conservators automatically and effectively extract crack information.

*5.2. Future Work*

According to the statistics generated from the experiments, the research methods and transfer learning ideas proposed in this paper have been proven to be effective. The models can only segment rammed earthen surface cracks, but the position, width, length, and depth of cracks are still uncertain, and thus, further optimization is needed. Moreover, the dataset of cracks on rammed earthen heritage sites is still small. More in-depth work is needed.

1. Since the image data of surface cracks in earthen heritage sites is still relatively small, there are few studies on the deep learning of surface cracks in earthen heritage sites. Therefore, it is very important to collect a large amount of image data of surface cracks on earthen heritage site surfaces and build a larger dataset. In the future, UAVs can be used to collect crack images, and image cropping and annotating can be performed using the trained model to establish a dataset.
2. By sharing the dataset of cracks on the surface of heritage sites and uploading crack pictures in real-time, a larger database of crack pictures was established, which is convenient for researchers to study the cracks.
3. This study focused on the identification of cracks. Future work can be done regarding the location, length, width, and depth of cracks.
4. The current solutions, which cannot be deployed in lightweight embedded systems, were limited due to the complexity of the network. For practical application, models that can be deployed based on UAVs will be able to detect cracks more efficiently.

**Author Contributions:** Conceptualization, Y.Z. and Z.Z.; methodology, Y.Z. and Z.Z.; software, Y.Z. and Q.L.; validation, W.Z.; data curation, W.Z. and Q.L.; writing—original draft preparation, Y.Z.; writing—review and editing, Y.Z. All authors have read and agreed to the published version of the manuscript.

**Funding:** This research was financially supported in part by the National Key Research and Development Program of China under Grant No. 2019YFC1520904.

**Institutional Review Board Statement:** Not applicable.

**Informed Consent Statement:** Not applicable.

**Data Availability Statement:** Not applicable.

**Conflicts of Interest:** The authors declare no conflict of interest.

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
