# Peer review of "Crack Segmentation on Earthen Heritage Site Surfaces"

_applsci, doi:10.3390/app122412830_

Round 1

Reviewer 1 Report

Abstract

It should be reviewed for readability in terms of English (verb tenses, prepositions, punctuation, etc). This is applied also to the rest of the document. The splitting of words (translineation) should also be reviewed.

An abstract should be clear even for someone not too close to the subject addressed, so a general review should be made with this focus, giving less methodological or operational details, and emphasizing the novelty of the study and the conclusions.

Introduction

The International/Chinese Framework of earthen heritage is not clear in the introduction.

The geographical context should be explicit if it is important for the study (I don’t think it is). Otherwise, an overview of the main characteristics of earthen heritage, building methods, constructive systems, and degradations processes would be more in line with the subsequent analysis of the crack in the surfaces.

Thus, the state of the art undoubtedly leads to the research question.

However, there is a lack of clarification of the possible universe of application, since earthen heritage sites include different typologies and building systems, from monolithic systems, such as rammed earth unplastered walls, to membrane systems, such as wattle and daub. This clarification and classification would be necessary right at the beginning of the introduction, as well as in the abstract.

A deeper analysis of the main characteristics of the cracking in pavements, concrete walls, and the different earthen surfaces would be important to assess in which cases the similarities in crack scale, crack sides and crack layout, will advise the use of those images in learning processes.

The separation, in a sub-chapter, of the literature review related to computer vision would also be advisable.

Materials and methods

The objective “This study proposed a deep learning method that can effectively detect and predict cracks for early monitoring and early warning of damage to earthen heritage sites” is not the focus of the article, that only deals with the detection part. In fact, in the conclusion, it is stated “This paper proposes a method for crack detection in earthen heritage sites”.

The article deals mainly with the evaluation of the reliability and precision of the application of different methods to cases of cracks in the earthen surfaces of Suoyang Ancient City.

From a methodological point of view, one important aspect is not explicit which is the real resolution of the pictures used both from reference cases and Suoyang Ancient City. I’m not talking about the picture size, (mentioned) but the survey detail (pixels in image correspondence to mm in reality).

Another aspect is the amount of data used for the deep learning processes and the real impact of the study with 1896 + 1124. + 348 images that all came from the same historical structure. This should be explicit.

Verification and Results

Tables are split on consecutive pages.

The parameters in table 3 should be explained.

Also, should be explained, “The optimizer is Adam…”

Conclusions and Future Work

The point “Advantages and Applicability”, in the text is more a resumé than an effective assessment of the advantages of the methods used in the concrete case in the study, mainly in answering the initial question – a way of effectively detecting cracks. How could it be used in real context by, for instance, a conservator?

With the revision of the aspects mentioned above, as well as a thorough revision in terms of citations, the article can be resubmitted as the subject address is very important and a step in implementing automated systems in early warning of risk in heritage sites.

Reviewer 2 Report

The paper addresses the issue of cracks in the main structures/surfaces of the earthen built heritage sites. This topic is particularly significant since the important heritage of earthen construction, located not only in Northwest China, but also in other Countries, stands as a remarkable testimony to the history and building techniques of places. Therefore, in light of this observation, and considering the urgency of efficiently predicting and monitoring the cracking of structures, the main goal of this research is to develop a method that can effectively detect cracks and their evolution in earthen heritage sites, so as to support and direct possible future maintenance works.

The paper, starting with a wide and thorough introduction, which well highlights the state of the art, constitutes a step of innovation and advancement in the scientific field, fitting within the journal's topics.

The literature references are several and mostly recent.

The paper's organisation is clear. However, the following suggestions are recommended:

1) Instead of briefly introducing the aims of the paper at the end of the introduction, it would be useful to dedicate a separate paragraph to them, so as to further clarify the main goal and sub-goals. In such a paragraph, it might be interesting to briefly anticipate the paper's structure, in order to illustrate its subsequent organisation.

2) In the Overwiew of the Materials and Methods paragraph (line 161), it would be useful to include a list of the methodological steps that are going to be illustrated and explored later. This could give greater clarity to this paragraph, which is considered particularly important.

3) It is also advisable to present the Results in a single paragraph, dividing it into several sub-paragraphs.

4) Finally, it is suggested to unify the Discussion section with the Conclusion one, since the discussion envisages future research developments (currently present in the Conclusion) and compares the achieved results with what has been done so far in the state of the art, placing them in a scientific context as wide as possible.

The paper, which has a good level of scientific accuracy, is interesting, making an effective contribution to the world of research. This contribution is reinforced by the development perspectives, which, as stated in the conclusions, are clear and valid.

The images, tables and graphs clearly and adequately illustrate the achieved results.
